# Research on health education and health promotion during the process of schistosomiasis elimination II Awareness among university students in endemic regions

**Jing Zhang**[☯], **Shuying Xie**[☯], **Huiqun Xie, Yifeng Li, Jun Ge, Junjiang Chen, Jun Wu, Fei Hu**[iD] *

Jiangxi Provincial Institute of Parasitic Diseases, Nanchang, Jiangxi, P.R. China

☯ These authors contributed equally to this work.
* hufei@21cn.com

## Abstract

In China, health education served as the primary method for controlling schistosomiasis and had significantly contributed to the management of schistosomiasis epidemics. In contrast, university students who lived/studied in schistosomiasis-endemic areas were often in the risk zone for schistosomiasis infection as part of their social practice and leisure activities. Thus, the risk of schistosomiasis transmission remained widespread and chronic. This study will conduct a survey and analyzed schistosomiasis awareness among university students in colleges and universities in endemic areas. The aim was to optimize intervention strategies once transmission had been interrupted. Students from two colleges and universities in the city of Gongqingcheng on Lake Poyang were selected and asked to complete a questionnaire via the online platform Questionnaire Star- The survey assessed exposure to snail-infested water, knowledge about schistosomiasis, and preferences for health education methods among students who had traveled to snail breeding areas. The survey took place from April 2 to April 4, 2024. The survey results showed that of the 4,847 respondents (49% male, 51% female), 53.8% reported exposure to snail-infested environments near their schools, and 38.4% had direct contact with snail-infested water. Of these, with those who dug for wild vegetables exposed significantly more nonendemic than endemic students to snail water (13.93% *vs.* 8.42%, $\chi^2 = 16.681$, $P = 0.000$). Awareness of schistosomiasis was low (31.08%), with limited knowledge about transmission (30.58%) and symptoms (42.91%). Of these, students from endemic areas were significantly more aware of transmission and symptoms than those from nonendemic areas (36.52% *vs.* 27.22%, $\chi^2 = 7.623$, $P = 0.006$ & 50.17% *vs.* 38.80%, $\chi^2 = 9.872$, $P = 0.002$). Preferred education methods included mobile multimedia (72.75%), brochures (68.68%), and physical promotional items (66.58%). In general, the overall awareness of schistosomiasis among university students remained low. There was a need to enhance health education in schools to improve disease prevention awareness within this population.

**Data Availability Statement:** All relevant data are in the manuscript and its supporting information files.

**Funding:** This work was supported by the Jiangxi Province Focus on Research and Development Plan, China (grant no. 20202BBGL73047 to FH) and the Science and Technology Plan of Jiangxi Provincial Health Commission, Chain (grant no. 202311184 to JZ). The funders had no role in the study design, data collection and analysis, decision to publish, or preparation of the manuscript.

**Competing interests:** The authors have declared that no competing interests exist.

## Author summary

Schistosomiasis remains a major public health problem in tropical and subtropical regions. Although China's schistosomiasis epidemic is at a very low level, the risk of infection still exists, and health education for schistosomiasis control in the new situation still has blind spots, especially for the entire health education for the mobile population has not yet been carried out. Among them, university students who live/study in schistosomiasis-endemic areas are often at risk of schistosomiasis infection due to social practices and leisure activities. In order to identify possible blind spots in schistosomiasis health education and promotion in the process of schistosomiasis elimination, we will conduct a survey and analyse the awareness of schistosomiasis among university students enrolled in colleges and universities in the infected areas. Our survey revealed a serious lack of knowledge about schistosomiasis control among university students. At the same time, it was found that, given the high level of internet access among university students, online health education can be used to raise awareness of schistosomiasis prevention among the university student population. This will provide a scientific basis for optimal intervention strategies for schistosomiasis health education and promotion following the interruption of transmission.

## Introduction

Schistosomiasis was a major zoonotic disease that posed a serious threat to human health and socioeconomic development [1]. Schistosomiasis remained a major public health problem in tropical and subtropical regions [2]. In China, schistosomiasis japonicum posed a significant threat to both humans and animals. The Chinese government had prioritized it as one of the three major infectious diseases for prevention and control [3]. The schistosomiasis control strategy implemented by the Chinese government through relentless efforts achieved remarkable results [4,5]. By the end of 2023, five provinces, namely, Shanghai, Zhejiang, Fujian, Guangdong and Guangxi, continued to meet the standard of schistosomiasis elimination, and seven provinces, namely, Jiangsu, Sichuan, Hubei, Yunnan, Jiangxi, Hunan and Anhui, met the standard of schistosomiasis transmission interruption [6]. Therefore, the national schistosomiasis epidemic was in a very low epidemic situation.

In the mid-1980s, the World Health Organization identified health education and health promotion as critical strategies for controlling and preventing schistosomiasis [7]. As one of the most cost-effective disease prevention strategies [8,9], health education on schistosomiasis was used as one of the main prevention and treatment tools in the World Bank-financed China Schistosomiasis Control Project implemented in China, with remarkable results [10–13], this made it an important part of the Chinese schistosomiasis control programme [14]. In particular, the recent Action Programme for Accelerating the Achievement of the Schistosomiasis Elimination Target (2023–2030) proposed the detailed implementation of "six actions" for schistosomiasis elimination, including health education promotion [15].

Throughout its efforts to control and prevent schistosomiasis, China had gained considerable expertise in health education and promotion, providing valuable technical assistance in the effort to eradicate the disease [14], but there were still shortcomings in the new situation; in particular, the whole process of health education for the migrant population had not yet been carried out [16].

At present, in China's schistosomiasis endemic areas, although the epidemiological situation of schistosomiasis had been further controlled, the intermediate hosts involved in the transmission of schistosomiasis had not been eliminated, and extensive and complex breeding environments still existed [17]. With the socioeconomic development of the countryside as well as the construction of towns and cities, some old snail breeding environments had developed into holiday and tourism areas, which were provided to the local/foreign population as tourism and recreation sites. Some of them had been developed and built as higher education institutions, such as the School of Basic Medical Sciences of Huazhong University of Science and Technology, Wannan Medical College (Riverside Campus), Gongqingcheng City University City, etc., and the scope of outgoing social practice and leisure activities of school students was in the risk zone of *Schistosoma* infection. Therefore, the risk of schistosomiasis transmission remained widespread and long-term [18,19].

University students were more vulnerable to mental health crises when faced with issues such as academic stress, relationships and punishment, and this vulnerability meant they needed more support and guidance to cope with the challenges [20], especially in schistosomiasis-endemic areas where they might face additional health and environmental burdens. Health education for university students is an important part of improving health for all [21]. As future leaders and professionals, the health awareness and behaviours of university students would have a long-term impact on society. In schistosomiasis-endemic areas, university students could raise community awareness of schistosomiasis and promote preventive measures through health education activities.

To explore sustainable development strategies for schistosomiasis control in the process of schistosomiasis elimination, a series of health education and health promotion studies were be conducted for different types of mobile populations. This study investigated and analyzed the awareness of schistosomiasis among university students in colleges and universities in endemic areas to provide a scientific basis for further improvements in postintervention strategies in China and other schistosomiasis-endemic countries.

## Methods

### Ethics statement

Our study protocol was approved by the Medical Ethics Committee of the Jiangxi Provincial Institute of Parasitic Disease (2024 No. 001). All methods and steps were performed in accordance with the relevant guidelines and regulations of the committee. Each participant was informed of the purpose, procedure and nature of the study on the first page of the questionnaire and was given the opportunity to withdraw at any time during the study. All participants provided both verbal consent prior to each interview. All participants were anonymous. Data anonymisation and pseudo-anonymisation techniques were used to protect the privacy of participants. This involved removing all personal identifiers from the data and replacing identifying details with artificial identifiers, which effectively reduced the risk of revealing participants' identities. At the same time, we ensured that data was stored using encrypted methods, whether on local servers or in the cloud. We also regularly backed up stored data in case a cyber event resultd in data loss.

### Participants

The Poyang Lake area was one of the areas with severe schistosomiasis epidemics in China, and we chose to conduct a survey along the lake in Gongqingcheng University city, which was less than 1.2 km from the nearest snail breeding environment. The city of Gongqingcheng University had been under construction since 2014, and the first batch of new students was

welcomed in autumn 2016. There were currently 14 colleges and universities with more than 110,000 students [22] (Fig 1). The formula for calculating the sample size for the required survey was $n = \frac{Z^2 \cdot p(1-p)}{E^2}$, where Z was the 95% confidence level (1.96), $p$ was the estimated total proportion (since it was unknown, it was conservatively taken as 0.5, which ensured that the variance was maximised at this point) and E was the maximum permitted error (3%). For this reason, the minimum sample size required for this survey was approximately 1062. Fig 2 showed the change in the number of respondents for the different contents of the questionnaire.

## Research methodology

Since the target population of this survey was university students, and considering the unpleasant and troublesome phenomenon of respondents in the survey process, the method of using voluntary respondents as the survey sample, i.e., the non-probability sample survey method, was adopted. The "Questionnaire Star"(https://www.wjx.cn/) platform was able to reach respondents quickly due to its efficient data collection and support for multiple distribution channels such as QR code, SMS, email, website embedding, etc. It was carried out by the respondents through the online platform "Questionnaire Star"; mandatory requirements were set for each entry to ensure the completeness of the questionnaire. The students interviewed were contacted by the counsellors of the institutions and procided with a link to the questionnaire of the platform, which they filled in on the condition of voluntary participation(they could refuse and would not be subjected to any unfair treatment). At the beginning, we randomly selected two universities to send the link to. After the respondents in the two universities completed the questionnaire, the link to the questionnaire was pushed to the students in the other 12 universities via WeChat (social software). The survey on the web platform started on 2 April 2024. By 4 April, the sample size of the survey had fully met our survey needs, so we blocked the survey instrument on the web platform.

## Research tools

The questionnaire was self-designed and developed in consultation with several schistosomiasis control and health education experts. It consisted of 4 multiple options could questions, 1 fill-in-the-blank question and 13 single options could questions. The questionnaire included basic information about the respondents, whether or not they had travelled to a snail environment around the school, the main time period of their activities, their knowledge about schistosomiasis and their acceptance of schistosomiasis health education and promotion methods (the full questionnaire is available in S1 Text). The questionnaire passed the reliability and validity tests with scores of 0.819 and 0.857 respectively.

## Definition of phrases

From an endemic area, the area where the student lived before coming to school is endemic for schistosomiasis.

From a nonendemic area, the area where the student lived before arriving at school is not a schistosomiasis-endemic area.

Snail water, is the natural habitat of the snail, such as the lake where the snail is found.

Field practices, refers to activities that are implemented or carried out in a real or practical setting, as opposed to theory.

Susceptible season, is a period of the year when the risk of transmission of schistosomiasis increases significantly due to a combination of the natural environment and human activities.

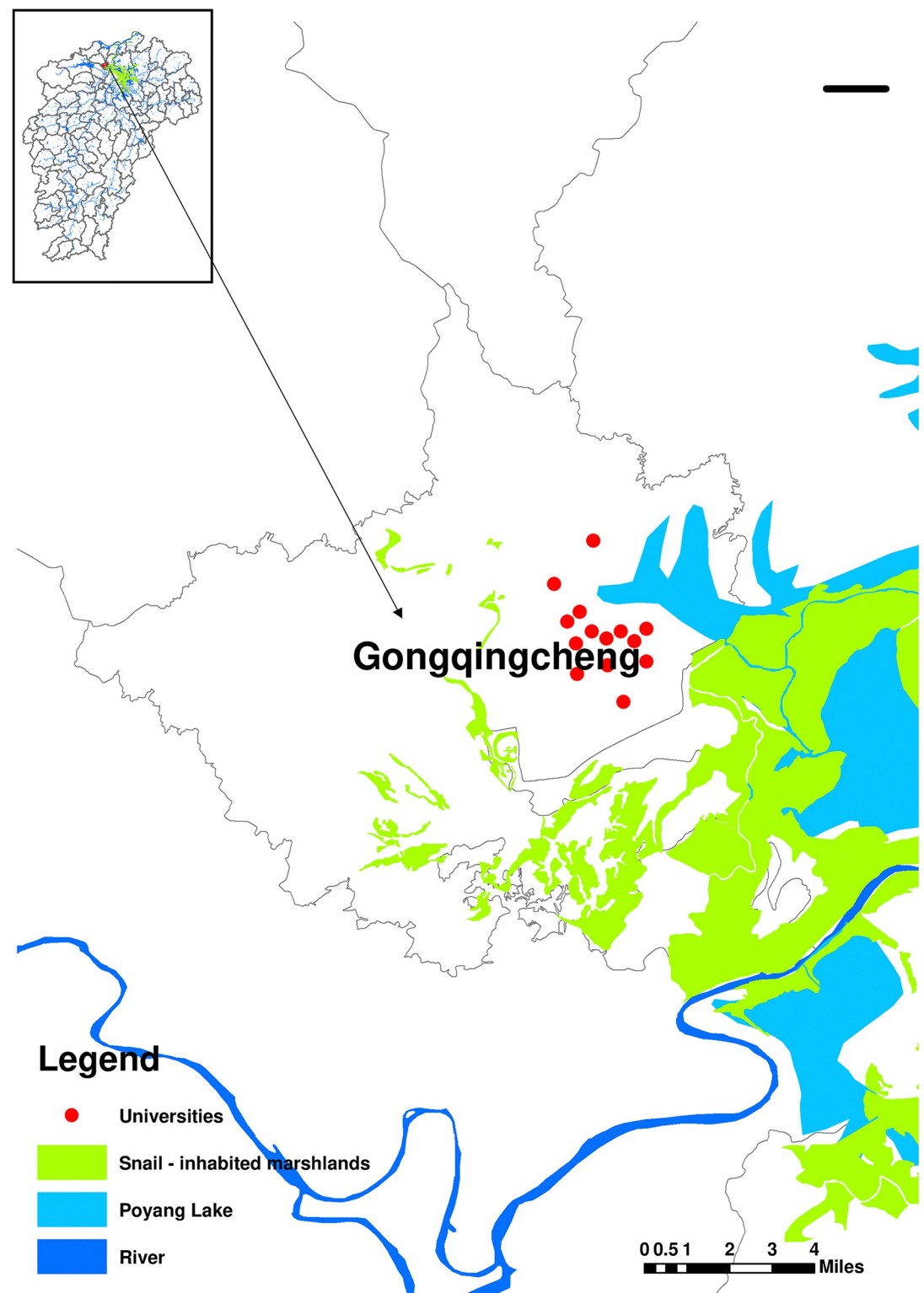

**Fig 1. Geospatial vector map of the area of the study.** (Vector map of administrative divisions of Jiangxi Province from the National Geomatics Centre of China, https://www.webmap.cn/, verification number: [GS (2024) 0650].)

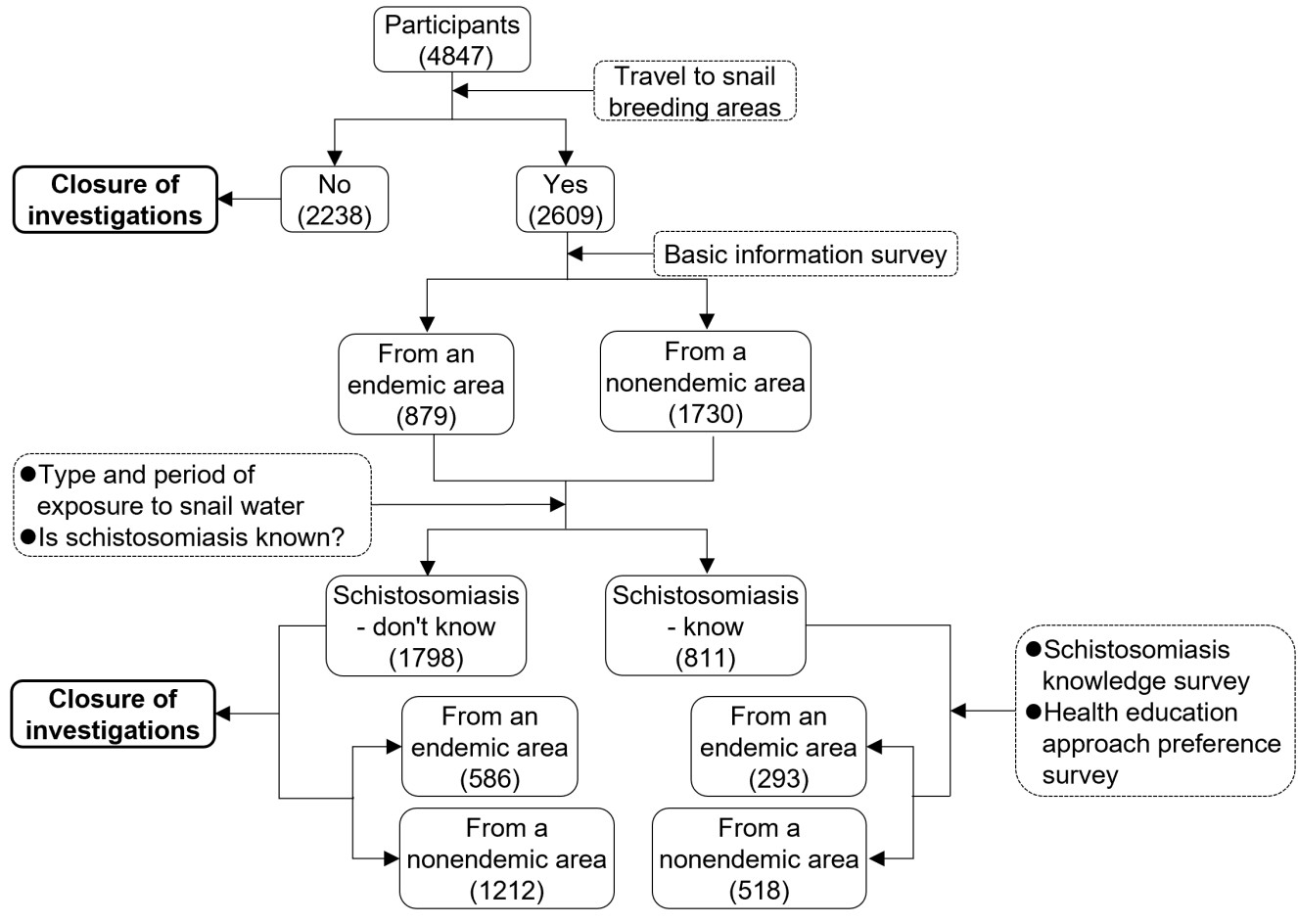

**Fig 2. Changes in the number of respondents.**

For schistosomiasis, the susceptible season is concentrated between April and October each year.

## Statistical analysis

Statistical analysis was performed via SPSS Statistics 27.0 software (SPSS Inc., Chicago, IL, USA), with a test level of $\alpha = 0.05$. General data and other relevant descriptions were described via descriptive statistics and compared via the $\chi^2$ test for comparison.

## Results

### Basic characteristics of the respondents

A total of 4,847 university students responded to the online questionnaire, including 2,367 males and 2,480 females; 99.53% (4,824/4,847) were in the 18–25 years age group, and 99.65% (4,830/4,847) were educated mainly at the tertiary or undergraduate level. Among these respondents, 53.83% (2609/4847) indicated that they had been in an area with a snail environment in the vicinity of their school. The results of the following analyses were based on this group of respondents (2609).

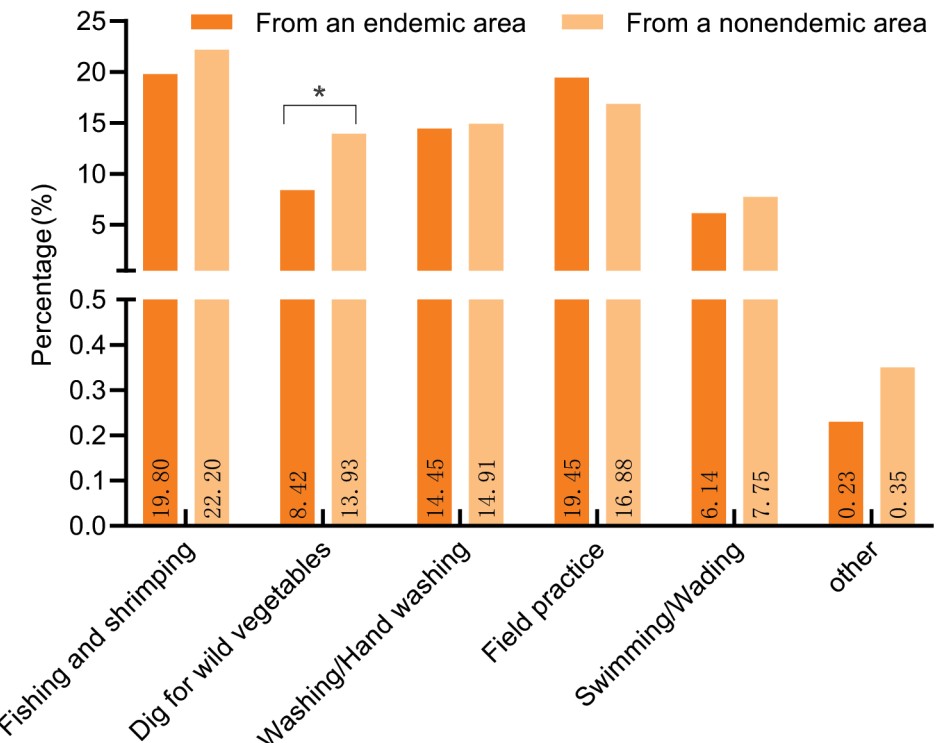

**Fig 3. Percentage of different types of exposure to snail-infested water.**

## Type and period of exposure to snail water

Overall, there was no significant difference in the percentage of students from endemic and nonendemic areas exposed to snail water (38.90% *vs.* 37.32%, $\chi^2 = 0.620$, $P = 0.431$). Regarding the different types of exposure to snail water, the percentage of university students exposed to snail water through field practices was greater for students from endemic areas than those from nonendemic areas (19.45% *vs.* 16.88%), but the difference between the two was not statistically significant ($\chi^2 = 2.648$, $P = 0.104$); all other types of exposure to snail water were greater among nonendemic than endemic students, with those who dug for wild vegetables exposing significantly more nonendemic than endemic students to snail water (13.93% *vs.* 8.42%, $\chi^2 = 16.681$, $P = 0.000$) (Fig 3).

We classified the periods of schistosome susceptibility as January–March, April–October, November–December and year round (irregular), with April–October being the susceptible season. From the periods of time that these university students were active in the area of snail-infested environments, there was no significant difference in the percentage of their activities in each period, and the overall percentage of exposure to snail-infested waters in each period was as follows: January–March < April–October < November–December < Year-round (irregular) (Fig 4).

## Knowledge of schistosomiasis

Only 31.08% (811/2609) of the 2609 university students were aware of schistosomiasis, with those from endemic and nonendemic areas accounting for 33.33% and 29.94% of their respective populations, respectively, which was not statistically significant when the two groups were compared ($\chi^2 = 3.129$, $P = 0.077$).

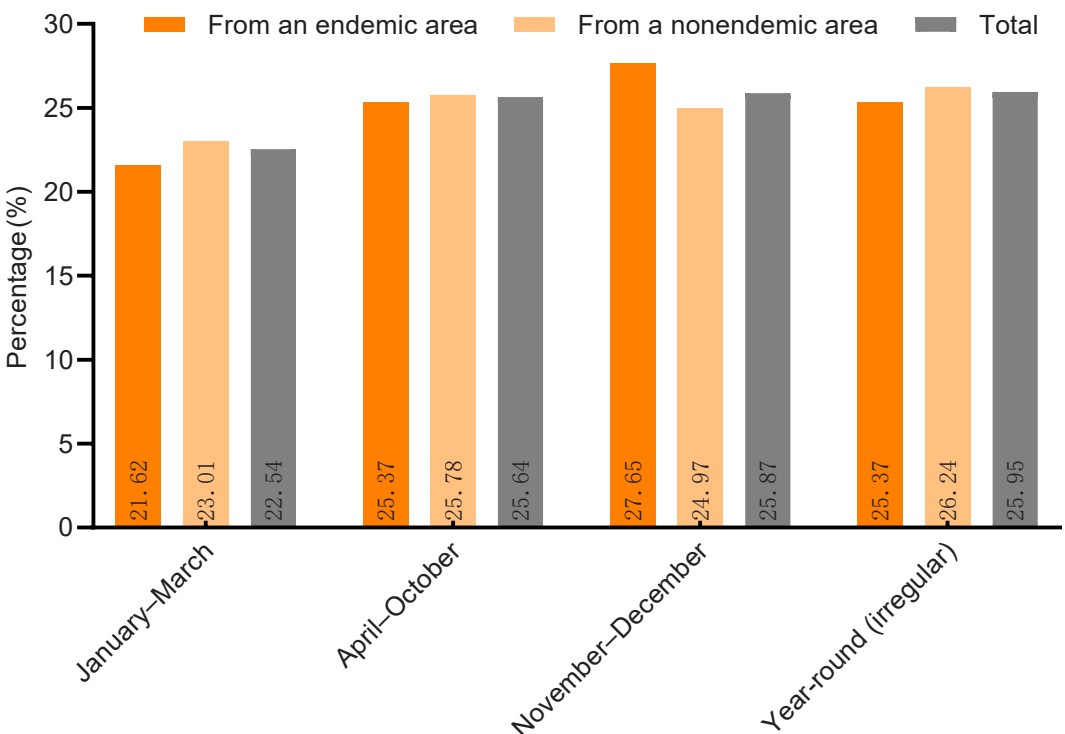

**Fig 4. Percentage of time spent by students from different regions in areas with snail environments.**

Of the 811 students who were aware of schistosomiasis only 30.58% correctly identified the mode of transmission, and significantly more students from endemic areas were correct (36.52% *vs.* 27.22%, $\chi^2$ = 7.623, *P* = 0.006). 49.20% of students were able to identify that the area they were travelling to was a schistosomiasis transmission area and only 30.33% were able to correctly identify the intermediate host of schistosomiasis transmission, there was no significant difference in the percentage of students from endemic and nonendemic areas who were correct on these two knowledge items (52.22% *vs.* 47.49% & 30.72% *vs.* 30.12%, $\chi^2$ = 1.674 & 0.032, *P* = 0.196 & 0.858). The percentage of correct answers to the question "what are the symptoms of schistosomiasis" was 42.91%, with 50.17% of students in endemic areas gettingit right, which was significantly higher than the 38.80% of students in nonendemic areas ($\chi^2$ = 9.872, *P* = 0.002) (Fig 5).

In addition, only 34.65% of those who "suspect schistosomiasis" would go for schistosomiasis testing at a specialized schistosomiasis control unit (CDC/Schistosomiasis Control Centre), whereas in China CDC/Schistosomiasis Control Centre was the only institution responsible for schistosomiasis control. The results of the survey revealed that 54.4% of university students had seen roadside warning signs or audio announcements about schistosomiasis control or had received leaflets about schistosomiasis control when traveling to schistosomiasis-affected areas.

## Mode of communication preferred by students for schistosomiasis health education

Among the interviewed students, with respect to their willingness to prefer schistosomiasis health education and publicity, the one that gained the highest recognition among the students was mobile multimedia (72.75%), followed by brochures, physical protection promotional

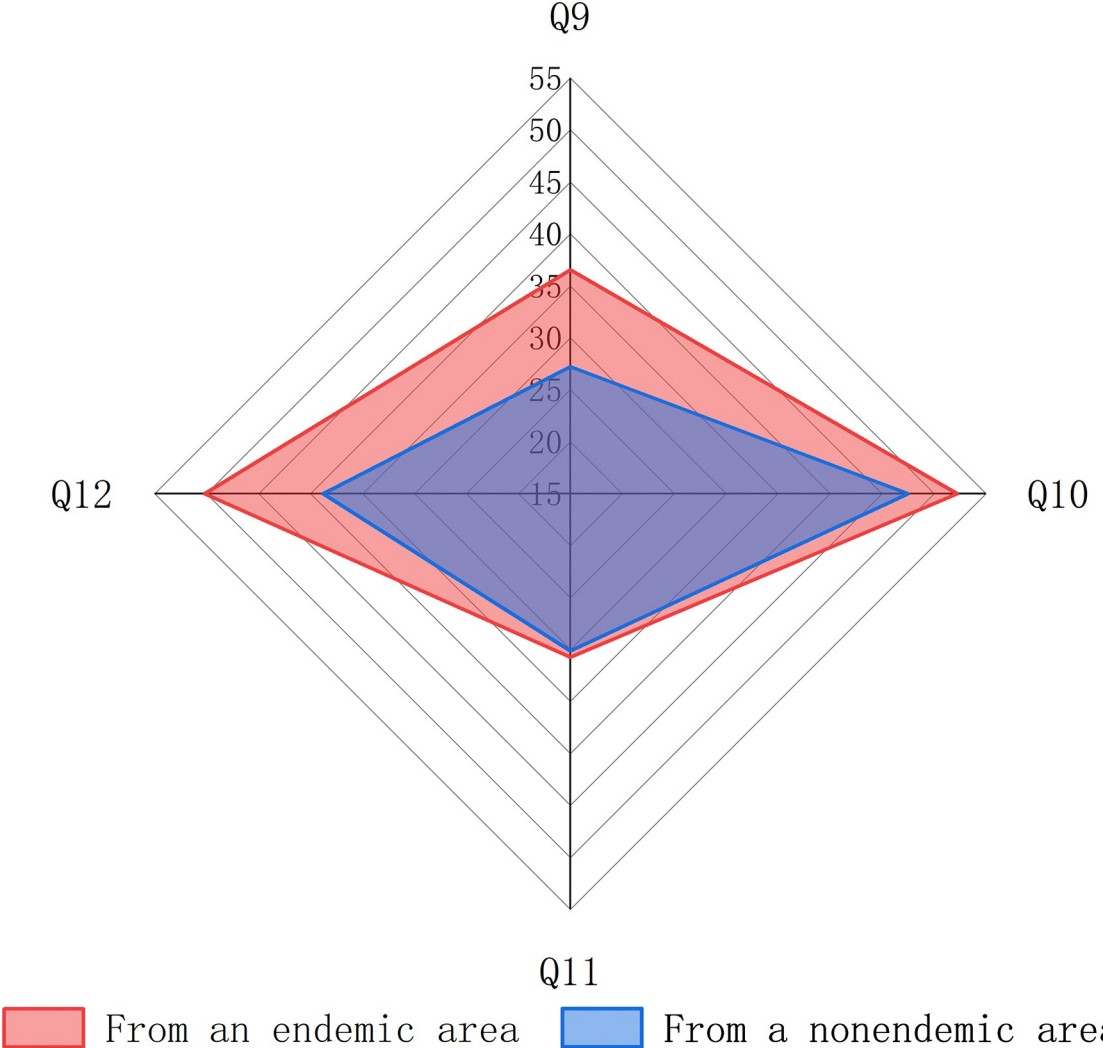

**Fig 5. Correctness of university students' answers on knowledge of schistosomiasis in different regions.** (Q9: how to contract schistosomiasis; Q10: students were aware that the areas of the lake they visited were transmission sites; Q11: which organisms transmit schistosomiasis; Q12: what are the symptoms of schistosomiasis).

items, the establishment of schistosomiasis warning signs in risk areas and the installation of intelligent audio warning systems in risk areas, with proportions of 68.68%, 66.58%, 66.21% and 54.62%, respectively. The recognition of mobile multimedia was significantly greater ($\chi^2 =$ 7.294, 8.167, 57.606; $P =$ 0.007, 0.004, 0.000) than the recognition of physical protective promotional items, the establishment of schistosomiasis warning signs in risk areas and the installation of intelligent audio warning systems in risk areas. Further analyses of preferences for different health education modalities among all categories of students from endemic and nonendemic areas showed no significant difference in preference rates between them ($\chi^2 =$ 0.965 & 2.446 & 0.001 & 0.384 & 1.765, $P =$ 0.326 & 0.118 & 0.979 & 0.536 & 0.184) (Fig 6).

## Discussion

Although the current prevalence of schistosomiasis in China had reached the transmission interruption standard, the widespread and complex distribution of the nail snail, the

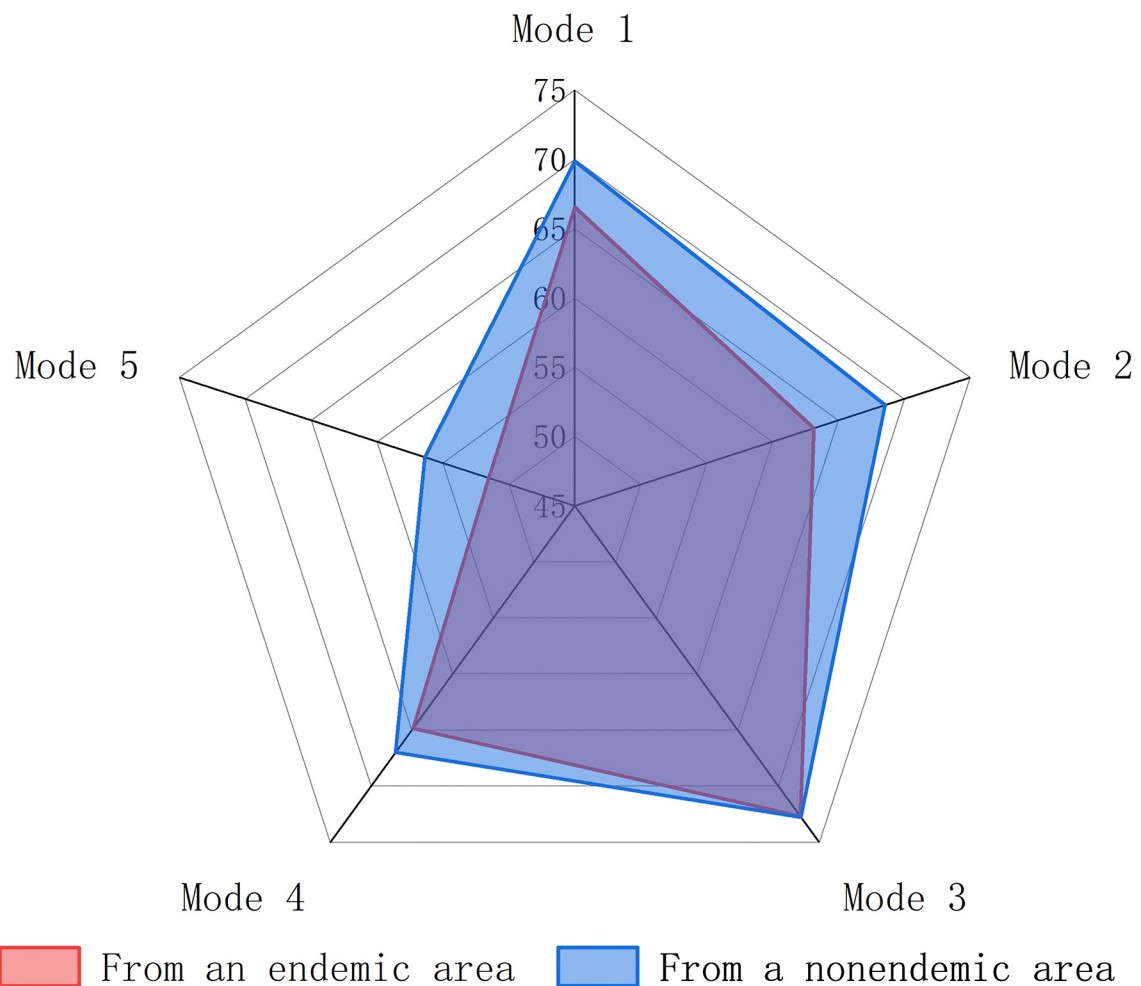

**Fig 6. Preferences for schistosomiasis health education promotion approaches among university students from different regions.** (Mode 1: Followed by brochures; Mode 2: Physical protective promotional items; Mode 3: Mobile multimedia; Mode 4: Establishing schistosomiasis warning signs in risk areas; Mode 5: Installing intelligent audio warning systems in risk areas).

intermediate host of *Schistosoma* japonicum, had posed some resistance to the task of complete schistosomiasis elimination by 2030. Therefore, postintervention strategies needed to be further improved and optimized [19].

Health education aimed to promote health and improve quality of life through the dissemination of knowledge and behavioral interventions that encourageed people to voluntarily adopt healthy behaviors and lifestyles and to eliminate or reduce health risk factors [11,12,23,24]. Most cases of schistosomiasis were closely related to the health knowledge and behavior of the population; health education activities aimed at disseminating health knowledge and promoting healthy behaviors played important roles in the prevention and control of schistosomiasis, and targeted health education could greatly reduce the risk of schistosomiasis infection and reinfection in the population [25]. To date, health education for schistosomiasis had focused on permanent residents and primary and secondary school students in endemic areas [9,11,16,26], and there had been is a lack of attention given to mobile populations that produced/lived in endemic areas [19].

Overall, there was a serious lack of knowledge about schistosomiasis among university students. Possible reasons included: 1) insufficient health education coverage, schistosomiasis education in endemic areas did not comprehensively cover the target population, resulting in insufficient awareness of the risk of schistosomiasis [27]; 2) similar behavioural habits, university students in endemic and nonendemic areas faced the same risk of schistosomiasis exposure during activities such as swimming and playing in water; 3) Similar environmental factors, natural water bodies such as lakes and rivers were present in endemic and nonendemic areas, which might lead to unintentional exposure to infected water; 4) Inadequate community participation, if community participation and implementation of schistosomiasis control measures were weak, it might lead to no significant difference in the exposure behaviour of university students in the two districts [28]. As a result, university students were at risk of contracting schistosomiasis while outdoors.

Schistosomiasis was a serious public health problem transmitted by contact with fresh water containing larvae, causing symptoms such as anaemia, diarrhoea and abdominal pain and, in severe cases, cirrhosis of the liver [29]. Knowledge of the symptoms was essential for early detection and treatment. The survey showed that knowledge of symptoms among university students in endemic areas (50.17%) was significantly higher than in nonendemic areas (38.80%), but knowledge of symptoms alone was not enough to effectively prevent transmission. Prevention of schistosomiasis required a combination of individual knowledge and community and policy support [30]. China had undertaken a large number of health education efforts on prevention and treatment, including the production of guidebooks and evaluation programmes [27]. These efforts had shown that education and public awareness could be effective in changing behaviour and reducing transmission. Therefore, in endemic areas, in addition to raising awareness of symptoms, there was a greater need to promote preventive behaviours through health education and community participation to more effectively control and prevent schistosomiasis transmission.

Currently, health education on schistosomiasis control was mainly targeted at primary and secondary school students, and there was no systematic health education intervention in higher education institutions in schistosomiasis endemic areas. This survey further confirmed the weak knowledge of schistosomiasis control among university students. Moreover, although the knowledge of schistosomiasis control among university students from endemic areas is greater than that among university students from nonendemic areas, it is only 36.52%, which might have been due to a lack of sustained reinforcement of schistosomiasis control knowledge after entering university, resulting in the dilution of the knowledge of schistosomiasis control over time [13,31]. In the future, health education on schistosomiasis for university students living in endemic areas should be strengthened to increase and sustain awareness of schistosomiasis control among the student population.

With the rapid development of data informatics, multimedia technologies such as the internet, television and mobile phones had become the main channels through which urban and rural residents obtained health information, communicated and interacted [32]. This time, among the university students surveyed who recognized schistosomiasis health education methods, the highest recognition was achieved by mobile multimedia (72.75%), which was significantly higher than that achieved by other health education methods. In the future, schistosomiasis control services should select health communication methods and approaches appropriate for different groups of people according to the needs of the target population and the relevant technical means and pay attention to the interactive and comprehensive use of new media and traditional health education communication methods to maximize the impact of health education and promotion [13,33].

### Limitations

This study had several limitations. Firstly, the sample was exclusively sourced from a single province in China where schistosomiasis was endemic, and it might not have reflect the overall picture. Secondly, we did not evaluate the knowledge of schistosomiasis prevention among university students who did not visited regions with snail populations; these students who did not visited areas with snail populations might have done so because they were aware of schistosomiasis transmission, and this might have affected the rate of correct knowledge of schistosomiasis among students. Additionally, there was a lack of information on the frequency of visits and whether the activity involved water contact (except of course swimming and washing). Nonetheless, our findings are currently preliminary, indicating that further research is essential to validate these results.

## Conclusion

The overall awareness of schistosomiasis among university students is not high, and there is a need to further strengthen school-based health education efforts, especially given the high internet accessibility of university students, and to actively conduct online health education to raise awareness of schistosomiasis prevention and treatment among the university student population. For this reason, health education-led risk control models for schistosomiasis should be strengthened after the interruption of schistosomiasis transmission [34].

## Supporting information

**S1 Text. List of contents of the questionnaire.**
(PDF)

**S1 Data. List of raw questionnaire data.**
(XLSX)

## Acknowledgments

We would like to thank the staff of the Gongqingcheng City Center for Disease Control and Prevention for their help with this study.

## Author Contributions

**Conceptualization:** Fei Hu.

**Formal analysis:** Jing Zhang, Shuying Xie.

**Funding acquisition:** Jing Zhang, Fei Hu.

**Investigation:** Jing Zhang, Shuying Xie, Huiqun Xie, Yifeng Li, Junjiang Chen, Jun Wu, Fei Hu.

**Methodology:** Jing Zhang.

**Visualization:** Shuying Xie, Jun Ge.

**Writing – original draft:** Jing Zhang, Shuying Xie.

**Writing – review & editing:** Yifeng Li, Jun Ge, Fei Hu.

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
