## [Decision Letter · Decision Letter 0]

24 Dec 2024

PNTD-D-24-01660

Research on health education and health promotion during the process of schistosomiasis elimination II A study examining the awareness of schistosomiasis among students at higher education institutions in endemic regions

Dear Dr. Hu,

Thank you for submitting your manuscript to PLOS Neglected Tropical Diseases. After careful consideration, we feel that it has merit but does not fully meet PLOS Neglected Tropical Diseases's publication criteria as it currently stands. Therefore, we invite you to submit a revised version of the manuscript that addresses the points raised during the review process.

Please submit your revised manuscript within 60 days Feb 22 2025 11:59PM. If you will need more time than this to complete your revisions, please reply to this message or contact the journal office at plosntds@plos.org. Please include the following items when submitting your revised manuscript:

We look forward to receiving your revised manuscript.

Kind regards,

Lynne Elson, PhD, MPH

Guest Editor

Qu Cheng

Section Editor

Shaden Kamhawi

co-Editor-in-Chief

Paul Brindley

co-Editor-in-Chief

**Additional Editor Comments:**

Raising awareness of schistosomiasis is a cornerstone of the strategy to control the disease in China. This manuscript is a descriptive report of an online schistosomiasis knowledge survey of university students in a new university town in an area endemic for S. japonicum. Respondents had come to the university town from endemic and non-endemic areas and some students reported having had contact with areas known to harbor the snail host. The knowledge of students originating from endemic and non-endemic areas are compared, but only for those who reported having visited sites with the snail host.

The three reviewers have quite different recommendations for the manuscript. On reviewing it myself I conclude that this piece of work will require substantive revision, including some re-analysis of data before it is acceptable. All comments from the three reviewers need to be addressed. Please note that Reviewer 1's comments are in an attachment.

The manuscript does not clearly describe the objectives, the methods that address them, nor the results that align with those.

The study groups are not well described or justified.

Some terms used are not well defined, such as snail water, endemic, exposure mode, exposure susceptible period. Why were these time periods chosen?

There is no description as to whether the presence of snails at the supposed exposure points was confirmed nor whether they were infected with S. japonicum.

A major shortfall is that the authors only analyzed a portion of the data, from those respondents who reported contact with "snail water" for their knowledge of schistosomiasis and yet those who did not were an important group to compare their knowledge as pointed out by Reviewer 2.

The choice of visualization method for the data is not ideal nor the description of the results.

The underlying data are not made available for the reviewers, nor the questionnaire.

The manuscript would be improved by including a map to show the study area.

I recommend the authors follow the STROBE checklist to ensure their manuscript describes their study according to international standards.

**Journal Requirements:**

1) Please provide an Author Summary. This should appear in your manuscript between the Abstract (if applicable) and the Introduction, and should be 150-200 words long. The aim should be to make your findings accessible to a wide audience that includes both scientists and non-scientists. Sample summaries can be found on our website under Submission Guidelines:

3) We note that your Data Availability Statement is currently as follows: "Correspondence and requests for materials should be addressed to F.H.". Please confirm at this time whether or not your submission contains all raw data required to replicate the results of your study. Authors must share the “minimal data set” for their submission. PLOS defines the minimal data set to consist of the data required to replicate all study findings reported in the article, as well as related metadata and methods (https://journals.plos.org/plosone/s/data-availability#loc-minimal-data-set-definition).

- The points extracted from images for analysis..

4) Please ensure that the funders and grant numbers match between the Financial Disclosure field and the Funding Information tab in your submission form. Note that the funders must be provided in the same order in both places as well. State the initials, alongside each funding source, of each author to receive each grant. For example: "This work was supported by the National Institutes of Health (####### to AM; ###### to CJ) and the National Science Foundation (###### to AM)." State what role the funders took in the study. If the funders had no role in your study, please state: "The funders had no role in study design, data collection and analysis, decision to publish, or preparation of the manuscript.".

If you did not receive any funding for this study, please simply state: u201cThe authors received no specific funding for this work.u201

**Reviewers' Comments:**

Reviewer's Responses to Questions

**Key Review Criteria Required for Acceptance?**

**Methods**

-Are the objectives of the study clearly articulated with a clear testable hypothesis stated?

-Is the study design appropriate to address the stated objectives?

-Is the population clearly described and appropriate for the hypothesis being tested?

-Is the sample size sufficient to ensure adequate power to address the hypothesis being tested?

-Were correct statistical analysis used to support conclusions?

-Are there concerns about ethical or regulatory requirements being met?

Reviewer #1: (No Response)

Reviewer #2: 1. There is no clear criterion for choosing the 2 out of the 14 institutions or colleges

2. Although it was mentioned that nonprobability sampling was used, the paper did not describe how participants were selected and recruited, and how they were given access to the questionnaire. This information is necessary to ensure that participation was indeed voluntary and that the manner of recruitment did not include any undue influence

3. RESULTS: "A total of 4,847 university students were interviewed..." were there interviews done or all participants answered the online questionnaire (self-administered)?

4. present the average or median age instead of using proportions since the age grouping used included almost all the study participants

5. It is unclear why the authors are comparing the exposure to snail water of students from endemic and non-endemic areas. Throughout the paper, the authors focused on comparing water contact exposure and knowledge of students from endemic and non-endemic areas prior to arriving in the university, when the aim tries to assess overall exposure and knowledge. If the significance of the study lies in determining whether there is a need to implement health education campaigns in universities in the area, there is really no value to testing the hypothesis whether students from endemic and non-endemic areas have differences in exposures and knowledge, since interventions will not make a distinction between these two types of students when implemented.

6. It was understandable that the questions on exposures to snail water or areas where there may be snail populations were only administered to those who went to areas where snails may be present. However, those who did not go to such areas should have been asked to answer the knowledge questions too. Excluding them from this part of the survey may constitute selection bias, and may introduce errors in the estimates of level of knowledge of the study population. Those students who refrained from going to areas where there may be snail populations may have done so because they have knowledge of schistosomiasis transmission.

7. what test was done to compare the recognition of mobile multimedia against other health education and publicity?

Reviewer #3: "This study will conduct a survey and analyze schistosomiasis awareness among university students (endemic population) in colleges and universities in endemic areas. The aim is to optimize intervention strategies once transmission has been interrupted." the objectives, sample size and study design are appropriate. Statistical analyses aligned to the objectives.

**Results**

-Does the analysis presented match the analysis plan?

-Are the results clearly and completely presented?

-Are the figures (Tables, Images) of sufficient quality for clarity?

Reviewer #1: (No Response)

Reviewer #2: Is there a way to ascertain that the students who went to the high risk areas were really exposed to snails? Because the way some parts of the discussions were written seem to indicate certainty of exposure (e.g. A total of 25.64% of the students were exposed to snails during the susceptible season from April--October, whereas 25.95% were exposed to snails throughout the year (irregular)). A visit to the area which are known to be habitats of the snail may be described as possible exposure instead of labeling them as exposed.

There was no clear description of how the visit to the snail area was asked: was it a visit in the past year? past 2 years? did the authors consider and collect data for multiple visits? Is there data on the number of students with multiple visits?

Reviewer #3: data analysis and results are well presented. The study aimed to optimize interventions and strategies once transmission has been interrupted. I was of the opinion that rather than just ask participants if they would accept schistosomiasis health education and publicity; the authors would outline the health education and health promotion strategies as well as put them to test to see which components of the interventions are effective and/or which need to be strengthened for maximum impact. this would allow them the opportunity to then analyze data to identify effective elements and refine the different health education and health promotion components accordingly.

the finding that "knowledge of schistosomiasis among the university students revealed that only

31.08% of them knew about the disease, and 30.58%, 49.20%, 30.33% and 42.91% of

these university students had no knowledge of "how to contract schistosomiasis", "how

to identify risk areas for infection", "which organisms transmit schistosomiasis" and

"what are the symptoms of schistosomiasis", was interesting. Since schistosomiasis health education as per China Action Programme to accelerate the achievement of the goal of eliminating schistosomiasis (2023--2030) and the Regulations on the Prevention and Treatment of Schistosomiasis. Why has health education on the disease leave out university students? Similarly, could there be recall bias on the part of participants? The study also does not state level of students who participated in the study; e.g. where they finalists or beginners? could there have been overlaps in the questions and/or responses of participants?

**Conclusions**

-Are the conclusions supported by the data presented?

-Are the limitations of analysis clearly described?

-Do the authors discuss how these data can be helpful to advance our understanding of the topic under study?

-Is public health relevance addressed?

Reviewer #1: (No Response)

Reviewer #2: No description of the limitations of the study and how these were addressed.

Reviewer #3: The authors make suggestions of the different communication approaches to be adopted for health education for different population groups but are not explicit about public health relevance. Limitations not clearly articulated.

**Editorial and Data Presentation Modifications?**

Reviewer #1: (No Response)

Reviewer #2: Table 1: what is under the subtotal column? the number and % exposed? then it should be labeled as that, because this column does not show the subtotal of the columns on the left.

Figure 1: The figure, based on the write up, is intended to show the proportion of students who visited the area on months when there is high susceptibility to exposure and infection. Maybe label the April to October as the susceptible months and the others as non-susceptible (can Jan-Mar and Nov-Dec be combined under this?). Also, please edit the title if the labels will be changed.

Reviewer #3: Perhaps the authors should be more explicit about the specific components of health education and health promotion strategies and interventions that were optimized in this study and describe the process. Just asking participants to state their willingness to accept health education and publicity, through mobile multimedia, brochures etc while it is important doesn't seem adequate if the goal is to optimize these strategies.

**Summary and General Comments**

Reviewer #1: (No Response)

Reviewer #2: The study has elucidated the significance of doing the survey in the specific study population. However, the direction of the analysis and discussion do not seem to address this significance very well. A comparison of the knowledge of those who visited and did not visit the high risk area may be more relevant in terms of planning for interventions. This can help determine whether people risk going to areas with snail populations because they have less knowledge on schistosomiasis as compared to those who avoided visiting those sites. If this is the case, then maybe an intervention to inform the students not only of schistosomiasis but of the risk that is present in the areas may be helpful.

Reviewer #3: Overall, there is scientific validity. The authors acknowledge the contagious and debilitating nature of schistosomiasis globally and specifically in China. the article is well written and presented in a scholarly manner. However, the authors must align their objective on optimization of health education strategies to their findings and conclusions. I think that the authors should also consider health education and health promotion for those communities in endemic areas for sustainable control and prevention of schistosomiasis.

PLOS authors have the option to publish the peer review history of their article (what does this mean?). If published, this will include your full peer review and any attached files.

Reviewer #1: No

Reviewer #2: **Yes: **Marianette T. Inobaya

Reviewer #3: **Yes: **Dr Nthabiseng A. Phaladze

**Figure resubmission:**
---

## [Decision Letter · Decision Letter 1]

19 Jan 2025

PNTD-D-24-01660R1

Research on health education and health promotion during the process of schistosomiasis elimination II Awareness among university students in endemic regions

Dear Dr. Hu,

Thank you for submitting your manuscript to PLOS Neglected Tropical Diseases. After careful consideration, we feel that it has merit but does not fully meet PLOS Neglected Tropical Diseases's publication criteria as it currently stands. Therefore, we invite you to submit a revised version of the manuscript that addresses the points raised during the review process.

Please submit your revised manuscript within 60 days Feb 18 2025 11:59PM. If you will need more time than this to complete your revisions, please reply to this message or contact the journal office at plosntds@plos.org. Please include the following items when submitting your revised manuscript:

We look forward to receiving your revised manuscript.

Kind regards,

Lynne Elson, PhD, MPH

Guest Editor

Qu Cheng

Section Editor

Shaden Kamhawi

co-Editor-in-Chief

Paul Brindley

co-Editor-in-Chief

**Additional Editor Comments:**

We are asking the authors to conduct another major revision since in their opinion, while many changes have been made in response to the reviewers, the Editors feel these are not satisfactory, particularly in light of the newly submitted questionnaire.

I have tried to make the requests clearer by being specific with page and line numbers and suggestions/ comments/ questions below.

Questionnaire

Thank you for adding the original and translated questionnaire and data in Supporting Information.

Some questions seem to be in the future tense including Q6 which asks “Which of the following activities will you engage in when you go to the Poyang Lake area?” This may be an accident of translation but needs to be changed if it was not asking about future visits to the lake.

Q11 seems to have two correct answers, Oncomelania and snails. How was this handled?

Q14 what is “blood-prevention”? This may be a translation issue. Maybe just use “prevention” ?

General

Much of the manuscript is written in the future tense. Manuscripts should always be in the past tense since they describe work that was conducted in the past.

Abstract

Make it clear that the knowledge questions were only answered by students that visited the lake area.

One of the key objectives and outcomes reported in the manuscript is comparing students from endemic and non-endemic areas and yet there is no mention of this in the abstract nor the key significant differences found and their implication for policy.

Methods

Thank you for adding a description of the sample size. Please add a statement about how long the survey instrument was available online. What was the trigger for it to be withdrawn/blocked? The sample size was calculated as 1062 and yet you report responses from 4847 students.

If following STROBE ideally you would include a flow chart of the student numbers at each stage: # approached by counsellors, # responded, # visited lake and responded to knowledge questions. Preferably split by endemicity of origin since that is your main comparator.

Line 164: this still refers to interviews. Did the counsellors interview the students or were they given a link to access the online questionnaire? This needs to be clear.

In the response letter there is an extensive justification of using the online platform but this does not appear in the manuscript. I recommend adding one sentence to justify the platform.

Page 7 line 168. Explain the process of selection of the first two universities.

Page 7 Line 173: many of your questions were multiple choice but in some only a single answer was allowed and in four of them multiple options could be selected. Describe this correctly.

line 174. Add to your new sentence “(the full questionnaire is available in Supporting Information S1)”.

Line 175: there is no question that asks the mode of travel to the lake. Change this sentence.

Page 7. You have a section with definitions. You have been asked to add to this some definitions for other phrases that you use e.g. “snail water”, “field practices”.

Results

Where p-values for the Chi2 tests are quoted they should be the actual p value, not just p<0.01 etc.

Page 8 line 213. There is nowhere in the questionnaire where you ask about duration of exposure. This heading and related text and Fig 3 caption should be changed. You only have the percentage of students who reported visiting the lake in specific time periods (groups of months) which you pre-set.

You could change the subtitle to : Type and period of exposure to snail water

Change Fig 3 caption to: Time of year students from different regions visited snail environments

Line 216-218: the percentage of university students exposed to snail water through field practices was greater in endemic areas than in nonendemic areas.

You did not assess this in non-endemic areas. Please be careful with your wording. This should be changed to read “the percentage of university students exposed to snail water through field practices was greater for students from endemic areas than those from nonendemic areas.”

Fig 2 & 3 Adding an * to the figure above the bars where p<0.05 comparing endemic & nonendemic students would help the reader.

Line 228. Please add a few words to explain why April-October is considered the “susceptible season”. Is this the rainy season when snail populations are highest?

Knowledge of schistosomiasis

The authors were asked to rewrite the results description to make it clearer which they have not done. They were also asked to explain Fig 4 better, which has not been done.

Line 244: The results of the schistosomiasis control knowledge test of 811 university students who were aware of schistosomiasis revealed that the knowledge rate of "how to contract schistosomiasis" was 30.58%, of which university students from endemic areas accounted for 36.52% of the population, which was significantly greater than that of 27.22% of the population from nonendemic areas (�2=7.623, P < 0.01).

This is a very long sentence and should be shortened. In addition, the word “control” should be deleted since the sentences refer to knowledge of transmission, not control.

Better wording would be: “Of the 811 students who were aware of schistosomiasis only 30.58% correctly identified the mode of transmission, and significantly more students from endemic areas were correct (36.5% v. 27.22%, p= 0.00x.)”

It is really not clear what the other sentences in this paragraph are stating. Which question do they refer to and was there any difference between students from endemic and non-endemic areas.

e.g. what does this statistic refer to? which was significantly higher than that of 38.80% of the population from nonendemic areas (�2 = 9.872, P < 0.01

Line 254: Since the reader is unlikely to know the correct answer to which facility treatment should be sought from, it is important to state that.

Fig 4 caption or labelling should make it more clear which question in the questionnaire is presented. Why not use the same question number? Make it clear what is presented is the % of students who gave the correct answer (I assume).

I don’t think “Knowledge 2” is “how to identify risk areas for infection”, but rather “students were aware that the areas of the lake they visited were transmission sites”. It should be correctly described based on the questionnaire.

Where are the results for questions 15, 16 and 17 which refer to prevention methods used? Why are these not presented?

Line 269: Acceptance of schistosomiasis health education and promotion methods

I suggest this subtitle should be edited to: Mode of communication preferred by students for schistosomiasis health education

Edit the accompanying text and Fig 5 caption accordingly as they were never asked if they would accept materials, but in what mode they would prefer.

What are “physical protection promotional items”? How are they different to brochures?

Line 279: how is this in contrast? The previous sentence compared modes of communication. Now you compare students from different areas.

Discussion

This largely repeats the Introduction and Results. There is no discussion of the results in relation to other similar surveys on schistosomiasis knowledge and practices in China nor elsewhere.

These were not addressed from Reviewer #1: While findings are summarized, the discussion lacks a deeper exploration of their implications. For example:

a. Why is knowledge of schistosomiasis so low among university students despite health education efforts in primary and secondary schools?

b. How might the identified preferences for multimedia education impact the scalability and sustainability of interventions?

One of the reviewers asked for more discussion of whether the higher level of knowledge of symptoms led to higher knowledge of appropriate health seeking or use of prevention methods. You added a paragraph, but it did not relate the results to each other nor to any previous publications.

Conclusion

Line 402: why recommend school-based prevention programs when you have shown students coming from schools in all areas of the country (? I assume) who have a low level of knowledge of schistosomiasis and prefer internet/social media-based communication? Why not just change this sentence to “internet-based education programs”?

Limitations

A short paragraph was added listing some limitations but no discussion of them and their impact on the outcomes.

Line 391: another limitation was that you have no way of knowing which universities respondents came from.

Additionally, the lack of information on the frequency of visits and whether the activity involved water contact (except of course swimming and washing).

**Comments to the Authors:**

**Please note that one of the reviews is uploaded as an attachment.**

**Reviewers' Comments:**

Reviewer's Responses to Questions

**Key Review Criteria Required for Acceptance?**

**Methods**

-Are the objectives of the study clearly articulated with a clear testable hypothesis stated?

-Is the study design appropriate to address the stated objectives?

-Is the population clearly described and appropriate for the hypothesis being tested?

-Is the sample size sufficient to ensure adequate power to address the hypothesis being tested?

-Were correct statistical analysis used to support conclusions?

-Are there concerns about ethical or regulatory requirements being met?

Reviewer #1: The requested revisions have been made.

**Results**

-Does the analysis presented match the analysis plan?

-Are the results clearly and completely presented?

-Are the figures (Tables, Images) of sufficient quality for clarity?

Reviewer #1: The corrections and queries were adequately addressed.

**Conclusions**

-Are the conclusions supported by the data presented?

-Are the limitations of analysis clearly described?

-Do the authors discuss how these data can be helpful to advance our understanding of the topic under study?

-Is public health relevance addressed?

Reviewer #1: The corrections on the limitations and other aspects of the conclusion have been addressed.

**Editorial and Data Presentation Modifications?**

Reviewer #1: Accept

**Summary and General Comments**

Reviewer #1: The quality of the manuscript has been greatly improved with the revision.

PLOS authors have the option to publish the peer review history of their article (what does this mean?). If published, this will include your full peer review and any attached files.

Reviewer #1: **Yes: **Abdulgafar Lekan Olawumi

**Figure resubmission:**
---

## [Editor Report · Decision Letter 2]

24 Jan 2025

Dear Prof. Hu,

We are pleased to inform you that your manuscript 'Research on health education and health promotion during the process of schistosomiasis elimination II Awareness among university students in endemic regions' has been provisionally accepted for publication in PLOS Neglected Tropical Diseases.

Best regards,

Lynne Elson, PhD, MPH

Guest Editor

Qu Cheng

Section Editor

Shaden Kamhawi

co-Editor-in-Chief

Paul Brindley

co-Editor-in-Chief

---

## [Editor Report · Acceptance letter]

2 Feb 2025

Dear Prof. Hu,

We are delighted to inform you that your manuscript, "Research on health education and health promotion during the process of schistosomiasis elimination II Awareness among university students in endemic regions," has been formally accepted for publication in PLOS Neglected Tropical Diseases.

Best regards,

Shaden Kamhawi

co-Editor-in-Chief

Paul Brindley

co-Editor-in-Chief
